# Comparative chromosome painting in *Spizaetus tyrannus* and *Gallus gallus* with the use of macro- and microchromosome probes

Carlos A. Carvalho[1,2], Ivanete O. Furo[2,3], Patricia C. M. O'Brien[4], Jorge Pereira[5], Rebeca E. O'Connor[6], Darren Griffin[6], Malcolm Ferguson-Smith[4], Edivaldo Herculano Corrêa de Oliveira [2,7] *

1 Programa de Pós-Graduação em Genética e Biologia Molecular, Universidade Federal do Pará, Belém, Pará, Brazil, 2 Laboratório de Citogenômica e Mutagênese Ambiental, SAMAM, Instituto Evandro Chagas, Ananindeua, Pará, Brazil, 3 Universidade Federal Rural da Amazônia (UFRA) Laboratório de Reprodução Animal (LABRAC), Parauapebas, Pará, Brazil, 4 Cambridge Resource Centre for Comparative Genomics, Cambridge, United Kingdom, 5 Animal and Veterinary Research Center, Universidade de Trá-os-Montes e Alto douro, Vila Real, Portugal, 6 School of Biosciences, University of Kent, Canterbury, United Kingdom, 7 Faculdade de Ciências Naturais, Instituto de Ciências Exatas e Naturais, Universidade Federal do Pará, Belém, Pará, Brazil

* ehco@ufpa.br, edivaldooliveira@iec.gov.br

**Data Availability Statement:** All relevant data are within the paper.

## Abstract

Although most birds show karyotypes with diploid number (2n) around 80, with few macro-chromosomes and many microchromosomes pairs, some groups, such as the Accipitri-formes, are characterized by a large karyotypic reorganization, which resulted in complements with low diploid numbers, and a smaller number of microchromosomal pairs when compared to other birds. Among Accipitriformes, the Accipitridae family is the most diverse and includes, among other subfamilies, the subfamily Aquilinae, composed of medium to large sized species. The Black-Hawk-Eagle (*Spizaetus tyrannus*-STY), found in South America, is a member of this subfamily. Available chromosome data for this species includes only conventional staining. Hence, in order to provide additional information on karyotype evolution process within this group, we performed comparative chromosome painting between *S. tyrannus* and *Gallus gallus* (GGA). Our results revealed that at least 29 fission-fusion events occurred in the STY karyotype, based on homology with GGA. Fissions occurred mainly in syntenic groups homologous to GGA1-GGA5. On the other hand, the majority of the microchromosomes were found fused to other chromosomal elements in STY, indicating these rearrangements played an important role in the reduction of the 2n to 68. Comparison with hybridization pattern of the Japanese-Mountain-Eagle (*Nisaetus nipalensis orientalis*), the only Aquilinae analyzed by comparative chromosome painting previously, did not reveal any synapomorphy that could represent a chromosome signature to this subfamily. Therefore, conclusions about karyotype evolution in Aquilinae require additional painting studies.

**Funding:** This research was partially funded by a grant to EHCO from CNPq (307382/2019-2) and to MAFS from the Wellcome Trust in support of the Cambridge Resource Centre for Comparative Genomics, by the Biotechnology and Biological Sciences Research Council (BB/K008161/1) to the University of Kent and by Fundação para a Ciência e a Tecnologia (UIDB/CVT/00772/2020) to JP. In addition, PROPESP/UFPA was responsible for financial support for the publication of this article. The funders (CNPq, Wellcome Trust, Biotechnology and Biological Sciences Research Council, Fundação para a Ciência e a Tecnologia and PROPESP/UFPA) had no role in study design, data collection and analysis, decision to publish, or preparation of the manuscript.

**Competing interests:** The authors have declared that no competing interests exist.

## Introduction

Usually, bird genome is organized in karyotypes consisting of few macrochromosomes and many tiny microchromosomes [1]. However, there are some exceptions. For instance, excluding the New World vultures (Cathartidae), which show similar karyotypes to the putative avian ancestral karyotype (PAK) with diploid number around 80, including 10 pairs of macrochromosomes and 30 pairs of microchromosomes [1], species belonging to the Order Accipitriformes present an interesting chromosomal diversity. They have lower diploid numbers, 2n, approximately = 54–68, and a reduction of microchromosomes to between 4 and 8 pairs, due mainly to fusions involving these small elements, occurred during their divergence [2–4].

In general, studies focusing on chromosome evolution in birds are based on comparative chromosome painting using chicken whole chromosome probes (*Gallus gallus*–GGA, 2n = 78), due to the similarity of the karyotype of this species with the PAK [5]. The use of this methodology in species of birds of prey has revealed that, despite the lower diploid numbers observed in this group, the large karyotype reorganization in Accipitriformes included multiple fissions in the macrochromosome pairs homologous to GGA1-GGA5. The reduction of the chromosome number would be due to the concomitant occurrence of several fusion events involving microchromosomes [6–11].

Microchromosomes are gene rich elements, and genome comparative analyses have shown their conservation as syntenic groups among distantly related bird groups [12, 13]. In fact, rearrangements involving microchromosomes were detected in few orders: Accipitriformes, Caprimulgiformes, Cuculiformes, Psittaciformes, and the Suliformes [13–15]. Due to difficulties of the isolation of individual microchromosome pairs by flow cytometry for specific probe production, most data concerning microchromosomes were obtained by the use of pools of microchromosomes, i.e., chromosome paints that recognize more than one pair. Therefore, improved identification of chromosome pairs involved in rearrangements is a priority if we are to achieve a more definitive analysis and identify synapomorphies based on chromosome characters [16, 17].

Currently, the order Accipitriformes is composed of four families, of which Accipitridae is the most diverse, with approximately 230 species distributed in 14 subfamilies [18]. Among them, the subfamily Aquilinae includes medium and large species, distributed globally, usually known as booted eagle. Usually, ten genera are found within Aquilinae. Cytogenetically, the only information concerning Aquilinae is the definition of the diploid number of six species (four genera), ranging from 2n = 66 to 82 [19].

The Black-Hawk-Eagle (*Spizaetus tyrannus*-STY) is a representative of this subfamily, found in South and Central Americas, from southern Mexico down to Argentina [18]. Considering that the only chromosomal analysis of *S. tyrannus* to date was based on conventional staining, revealing a karyotype within the Aquilinae standard, with 2n = 68 [1], the aim of this study was to present the cytogenetic mapping of *S. tyrannus* by comparative painting. In addition to whole-chromosome paints of *Gallus gallus* (GGA), we used BAC probes from GGA clones that identified 11 individual pairs of microchromosomes. The results were compared to *Nisaetus nipalensis orientalis*-NNI (2n = 66) [10], also from the subfamily Aquilinae, in order to identify chromosomal rearrangements related to karyotype evolution in this group.

## Results

### Karyotype description

The karyotype of *Spizaetus tyrannus* presented 2n = 68, consisting of 21 meta-submetacentric pairs (pairs 1–4, 6–7, 9–10, 12, 14, 16–17, 19–22, 24–29 and the sex chromosomes, Z and W), seven acrocentric (pairs 5, 8, 11, 13, 15, 18 and 23), and four pairs of microchromosomes

(pairs 30–33). The Z chromosome is a large metacentric, with size between pairs 3 or 4, while the W chromosome is an average submetacentric, similar in size to pairs 8 or 9 (Fig 1). In Table 1, we reported some differences in chromosome morphology of *S. tyrannus* described by Tagliarini et al., [1].

### Comparative chromosome painting

*Gallus gallus* probes used in the fluorescent *in situ* hybridization (FISH) experiments produced reproducible results. Hybridizations with chromosome-specific probes for the first ten pairs of GGA produced 22 signals, with the first five pairs producing multiple signals, ranging from 2 to 6 number (Fig 2). For instance, GGA1 probe painted six distinct pairs in the *S. tyrannus* karyotype (pairs 5, 6, 12, 14, 18, and 25), while the probes GGA6-10 pairs showed only one signal each. Table 2 details the distribution of the signals produced by GGA whole specific probes in the karyotype of *S. tyrannus*.

A total of 19 out of 22 *G. gallus* BAC clones produced results. Both BACs from the GGA22 chromosome did not produce any detectable signal, as well as one of the BACs from GGA21. Among the 19 probes that gave good quality results, both proximal (BACp) and distal (BACd) referring to 8 pairs, were found in the same segment in the STY karyotype. However, BACs corresponding to GGA17 hybridized to two different pairs—BAC17p marked STY 9q, while BAC17d marked STY 24q. (Fig 3). All Chicken BACs and their respective homology in the karyotype of *S. tyrannus* are summarized in Table 3.

Homologies obtained both by whole chromosome painting and BAC probes are shown in Fig 4.

### Discussion

The karyotype of *S. tyrannus* obtained herein presented 2n = 68, confirming data from a previous report [1]. We report slight differences in chromosome morphology however, due to the higher number of biarmed pairs (Table 1).

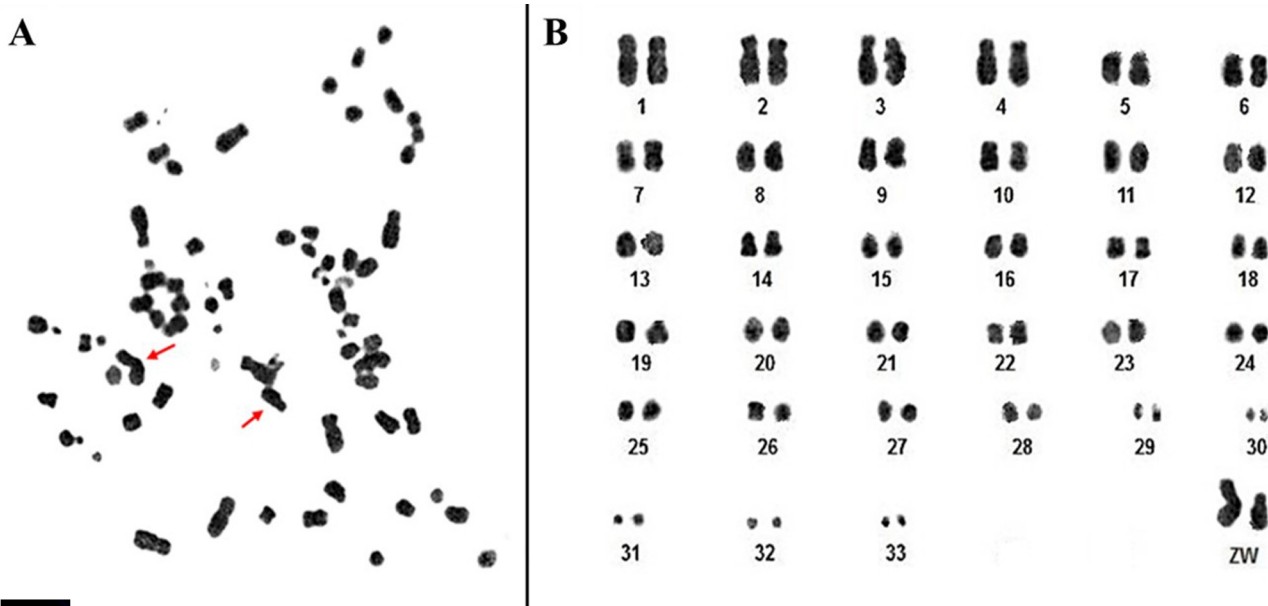

**Fig 1. Metaphase (A) and karyotype (B) of *S. tyrannus* with 2n = 68, obtained with Giemsa conventional staining.** The red arrows in (A) indicate the sex chromosomes. Scale bar: 5μm.

**Table 1. Karyotype of *S. tyrannus* described by Tagliarini et al. [1] and at this study.**

| Pairs | This study | [1] | Pairs | This study | [1] |
|---|---|---|---|---|---|
| **Chr 1**. | SM | SM | Chr 18. | AC | AC |
| **Chr 2**. | SM | SM | Chr 19. | SM | SM |
| **Chr 3**. | SM | SM | Chr 20. | SM | SM |
| **Chr 4**. | SM | SM | Chr 21. | SM | SM |
| **Chr 5**. | AC | SM | Chr 22. | SM | SM |
| **Chr 6**. | SM | ST | Chr 23. | AC | AC |
| **Chr 7**. | SM | SM | Chr 24. | SM | AC |
| **Chr 8**. | AC | ST | Chr 25. | SM | AC |
| **Chr 9**. | SM | SM | Chr 26. | SM | AC |
| **Chr 10**. | SM | ST | Chr 27. | SM | SM |
| **Chr 11**. | AC | SM | Chr 28. | SM | AC |
| **Chr 12**. | SM | SM | Chr 29. | SM | SM |
| **Chr 13**. | AC | SM | Chr 30. | Micro | Micro |
| **Chr 14**. | SM | AC | Chr 31. | Micro | Micro |
| **Chr 15**. | AC | AC | Chr 32. | Micro | Micro |
| **Chr 16**. | SM | ST | Chr 33. | Micro | Micro |
| **Chr 17**. | SM | ST | Chr ZW. | M and SM | M and SM |

(Metacentric: M; Submetacentric: SM; Subtelocentric: ST; Acrocentric: AC).

The results of comparative chromosome painting with whole chromosome probes of *G. gallus* showed a similar pattern to other birds of prey in the family Accipitridae, with a large reorganization of the syntenic groups homologous to the first five pairs of *G. gallus*. That is, each probe (GGA1—GGA5) corresponded to at least two distinct pairs (Fig 3). The most extreme examples are the fission of GGA1 into six pairs in STY, and GGA3 into four distinct pairs. These results are congruent with other birds of prey, considering that GGA1 can reveal syntenic segments in four pairs (*Gypaetus barbatus*, 2n = 60) to seven pairs (*Nisaetus nipalensis orientalis*—NNI), while GGA3 is hybridized to four pairs in all species analyzed in this family. The exception is NNI where it hybridizes to only 2 pairs [11]. On the other hand, GGA6—GGA10 are conserved syntenies, with only one signal for each pair.

All associations observed in the karyotype of STY based in its homology with *G. gallus* are represented in Fig 4. In general, 16 fissions and 13 fusions were detected, totalizing 29 rearrangements in the karyotype of STY when compared to *G. gallus*, with fissions occurring mainly in relation to the first five pairs of macrochromosomes and fusions involving mainly the microchromosomes. In the microchromosomes, chicken BAC probes showed that their syntenies were not disrupted by fission events as probes for proximal and distal regions were found hybridizing to the same pair in STY, except for GGA17, which produced signals in STY9 and STY24. However, all the identified BAC signals showed that each GGA microchromosome was fused to a STY segment homologous to either a GGA macro or microchromosome. This indicates that chromosomal fusions played an important role in reducing the diploid number in STY and other Accipitriformes. It is important to note that not all GGA microchromosomes are represented by chicken BACs, and hence other fusions must have occurred in this species to maintain 2n = 68.

The closest subspecies to *Spizaetus tyrannus* with chromosome painting data is the NNI, with 2n = 66 [10]. Although geographically separated, they are morphologically similar, and until the last decade were classified as part of the same genus. Despite now being separated

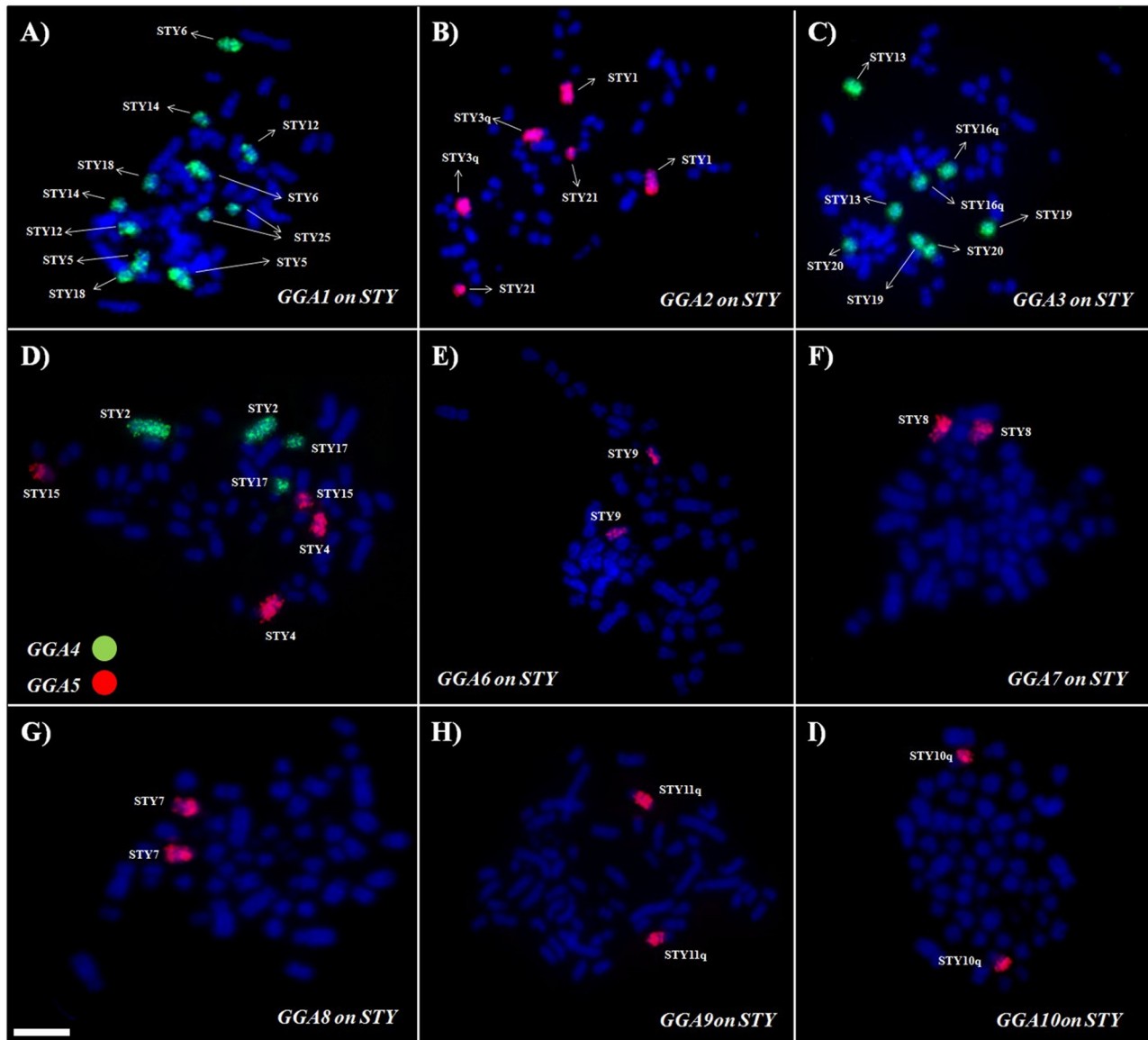

**Fig 2. Representative results of FISH experiments using *G. gallus* chromosome-specific probes corresponding to pairs GGA1 to GGA5 in *S. tyrannus* karyotype.** Red and green signals represent probes labelled with Cy3 or FITC, respectively. Scale bar: 5μm.

**Table 2. Results of hybridizations with *G. gallus* probes showing the homology between GGA probes in the karyotype of *S. tyrannus* (STY).**

| Probes | STY Chromosomes | Probes | STY Chromosomes |
|--------|-----------------|--------|-----------------|
| GGA1 | (5, 6, 12, 14, 18, 25) | GGA6 | 9 |
| GGA2 | (1, 3q, 21) | GGA7 | 8 |
| GGA3 | (13, 16q, 19, 20) | GGA8 | 7 |
| GGA4 | (2, 17) | GGA9 | 11q |
| GGA5 | (4, 15q) | GGA10 | 10q |

(q = long arm).

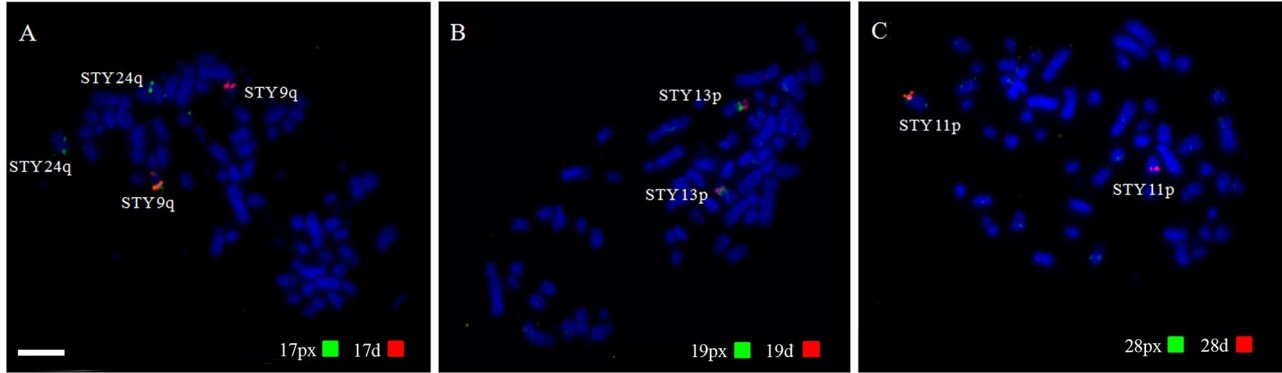

**Fig 3. Representative results of hybridizations with some *G. gallus* BACs probes in the karyotype of *S. tyrannus*. (A1 and A1.1) chicken BAC17 was the only one to hybridize to different chromosomes.** Red signals represent probes labeled with Cy3, corresponding to the proximal region (px); Green signals represent probes labeled with FITC, corresponding to the distal region (d). Arrows indicate the signals. Scale bar: 5μm.

**Table 3. Summary of the results of experiments using GGA BACs on the karyotype of *S. tyrannus*** (* = BACs marking the same segment in STY karyotype; px = proximal region; d = distal region; p = short arm; q = long arm).

| GGA | BAC ID | STY | GGA | BAC ID | STY |
|---|---|---|---|---|---|
| 17px | CH261-113A7 | 9q | 23d | CH261-90K11 | 23p* |
| 17d | CH261-42P16 | 24q | 24px | CH261-103F4 | 15p* |
| 18px | CH261-60N6 | 19p* | 24d | CH261-65O4 | 15p* |
| 18d | CH261-72B18 | 19p* | 25px | CH261-59C21 | 20p* |
| 19px | CH261-10F1 | 13p* | 25d | CH261-127K7 | 20p* |
| 19d | CH261-50H12 | 13p* | 26px | CH261-186M13 | 27p* |
| 21px | CH261-83I20 | No signal | 26d | CH261-170L23 | 27p* |
| 21d | CH261-122K8 | 4p | 27px | CH261-66M16 | 16p* |
| 22px | CH261-40J9 | No signal | 27d | CH261-28L10 | 16p* |
| 22d | CH261-18G17 | No signal | 28px | CH261-64A15 | 11p* |
| 23px | CH261-191G17 | 23p* | 28d | CH261-72A10 | 11p* |

into distinct genera, molecular data support their close phylogenetic relationship [20]. Nevertheless, the comparative chromosome painting detects many differences. For instance, GGA1-9 probes produced signals in 21 pairs in STY, and 22 in NNI; the difference was due to an extra fission of GGA1 in NNI. Despite both species presenting three fusions involving the first nine pairs with microchromosomes (STY: pairs 4, 7 and 9; NNI: pairs 2, 4 and 9), none of them share the same GGA syntenic groups, and microchromosomes involved in NNI were not identified. Additionally, in both species GGA3 hybridizes to 4 pairs, however in STY all these segments are fused with microchromosomes (GGA: pairs 18, 19, 24 and 25), whereas in NNI only one segment of GGA3 is fused with a microchromosome (unidentified pair) [10].

Regarding the phylogenetic relationship of Aquilinae with other subfamilies within Accipitridae, although STY and NNI present some karyotypic similarities common to diurnal birds of prey, such as recurrent breakpoints mainly in relation to the GGA1-GGA5 pairs [10, 11], we did not identify any synapomorphic associations which could represent ancestral characteristics for the Aquilinae [21, 22]. Hence, while other subfamilies, such as Buteoninae and Harpiinae present well-established chromosomal signatures that allow the elaboration of their putative ancestral karyotypes [7], the available chromosome data indicate the absence of

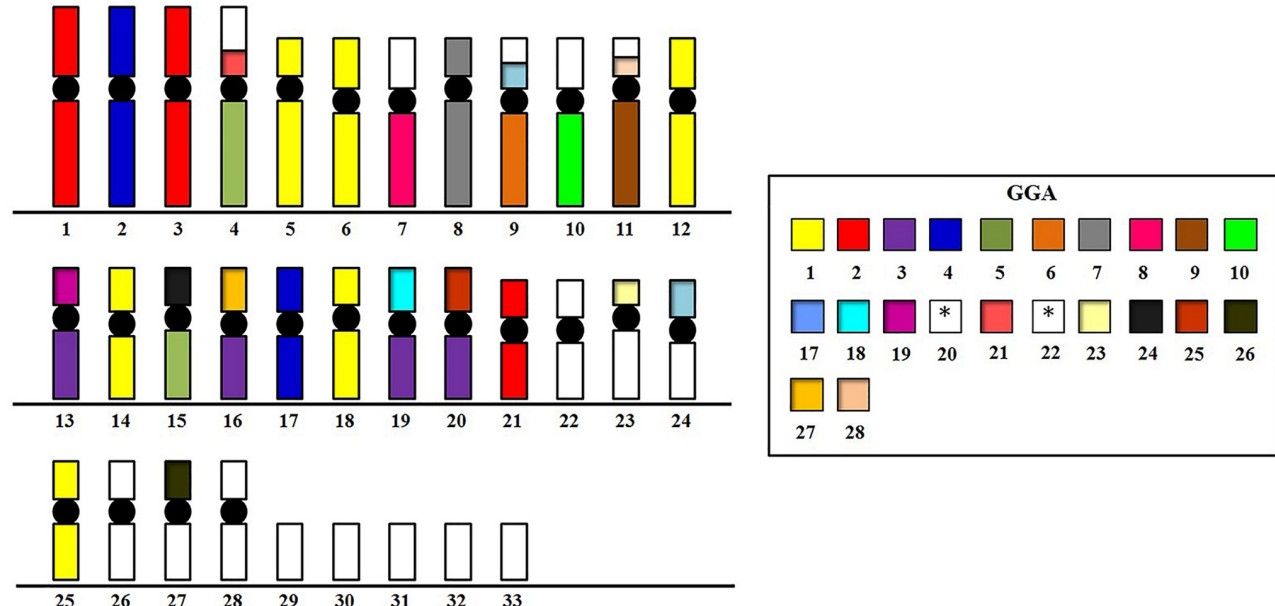

**Fig 4. Idiogram representing the homology between the *S. tyrannus* chromosomes and the macrochromosome chromosome-specific probes and microchromosomes BAC clones from *G. gallus*.** Empty boxes mean no signal detected in those chromosomes with the set of probes used. BACs corresponding to pairs 20 and 22 (*) was not used or did not produce any detectable signals, respectively.

chromosomal signatures between STY and NNI, which can be explained by their significant geographic isolation, inhabiting opposite regions in the globe.

## Conclusion

The present work is the first comparative chromosome mapping of a species in the genus *Spizaetus*, *S. tyrannus*, and has revealed substantial karyotypic reorganization common to birds of prey of the family Accipitridae. Together with *G. gallus* chromosome-specific probes for the larger pairs, chicken BACs were able to provide a more comprehensive result with additional information on the organization of the *S. tyrannus* karyotype. There are many similarities with the *N. nipalensis orientalis*, including numerous fissions of the first five pairs homologous to GGA with only one less in STY (21 events against 22 in NNI), and three fusions involving homologues of GGA1-GGA9 chromosomes and microchromosomes, but with breakpoints that are not shared between these two species. For a broader analysis at the phylogenetic level, it would be necessary to have comparative mapping of other species of the genus *Spizaetus* so that an ancestral karyotype of this genus could be suggested.

## Methods

### Samples and chromosome preparations

The experiments followed the standards approved by the Ethics Committee for the Use of Animals in Research (CEPAE-UFPA under number 170–13). We performed fibroblast cell cultures from skin biopsies and feather pulp of *Spizaetus tyrannus* (STY) obtained from two female individuals maintained in Zoos (Criadouro Gavião Real, Capitão Poço, Brazil), following the protocol of Sasaki et al. [23] with modifications. After tissue cleavage in Petri dishes, the samples were incubated with 1% type 1 collagenase (GIBCO) for 1 hour at 37˚C for tissue dissociation. Metaphase chromosomes were obtained after incubation for one hour with

Colcemid (0.05 μg/mL), hypotonic solution (KCl at 0.075 M) for 20 minutes and fixation in methanol/acetic acid (3:1). Karyotype analysis was performed using conventional staining with 5% Giemsa in 0.07 M phosphate buffer (pH 6.8) for 5 minutes, slides were analyzed using a 100× objective (Leica, CO, USA) and GenASIs software (ADS Biotec, Omaha, NE, USA).

## GGA probes and FISH experiments

Two types of *Gallus gallus* probes were used: whole-chromosome-specific probes of the first 10 pairs, and bacterial artificial chromosomes (BACs) probes from 11 microchromosome pairs. Whole chromosome paints were developed and provided by the Cambridge Resource Center for Comparative Genomics (Cambridge, UK) using the Fluorescent Activated Cell Sorting (FACS) technique and labeled with biotin, fluorescein and/or digoxigenin (Roche Diagnostics, Mannheim, Germany), and detected with the addition of avidin-Cy3 (or Cy5) or anti-digoxigenin-rhodamin (Vector Laboratories, Burlingame, CA, USA). BAC clones ranged from 150,000 kb to 210,000 kb in size were selected from the CHORI-261 Chicken BAC library (Children's Hospital Oakland Research Institute, Oakland-USA), corresponding to sequences from the proximal and distal regions of the microchromosomes (each pair represented by two BACs, in a total of 22 BACs covering pairs 17 to 28, except for pair 20). Clones were produced following the protocol of the mini prep kit (Qiagen, Hilden, Germany) and labelled directly by fluorescein isothiocyanate (FITC) (green) or Texas Red (red) through Nick Translation (Roche, Mannheim, Germany).

Hybridization experiments followed standard procedures [7, 12]. Probes (1 μL labelled probe in 14 μL hybridization buffer) were denatured at 70ºC for 10 min and preannealed for 30 min at 37ºC. Hybridization mix was added on slides with chromosome preparations previously denatured at 70% formamide for 1 min and 20 s and dehydrated by serial ethanol dehydration (70%, 90% and 100%). Detection was performed using Avidin-Cy5 or anti-digoxygenin (Vector Laboratories, Burlingame, CA, USA). Slides were analyzed with an Olympus BX-61 epifluorescence microscope equipped with a cooled CCD camera and appropriate filters. Images were captured using SmartCapture3 (Digital Scientific UK).

## Acknowledgments

We would like to thank all the staff of the Laboratório de Citogenômica e Mutagênese Ambiental (SAMAM, IEC) and Cambridge Resource Centre for Comparative Genomics for their technical support.

**Ethics approval**: The experiments were carried out according to the ethical protocols approved by an ethics committee (CEUA—Federal University of Pará) under no. 170/2013 and SISBIO 68443–1.

## Author Contributions

**Conceptualization:** Carlos A. Carvalho, Edivaldo Herculano Corrêa de Oliveira.

**Formal analysis:** Carlos A. Carvalho, Ivanete O. Furo.

**Funding acquisition:** Darren Griffin, Malcolm Ferguson-Smith, Edivaldo Herculano Corrêa de Oliveira.

**Investigation:** Carlos A. Carvalho, Ivanete O. Furo, Darren Griffin.

**Methodology:** Ivanete O. Furo, Patricia C. M. O'Brien, Jorge Pereira, Rebeca E. O'Connor.

**Supervision:** Darren Griffin, Malcolm Ferguson-Smith, Edivaldo Herculano Corrêa de Oliveira.

**Validation:** Rebeca E. O'Connor.

**Visualization:** Rebeca E. O'Connor.

**Writing – original draft:** Carlos A. Carvalho.

**Writing – review & editing:** Patricia C. M. O'Brien, Darren Griffin, Malcolm Ferguson-Smith, Edivaldo Herculano Corrêa de Oliveira.

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
