## [Decision Letter · Decision Letter 0]

19 Aug 2021

PONE-D-21-20639

Comparative Chromosome Painting in the Black-Hawk-Eagle (Spizaetus tyrannus) and Gallus gallus with the use of Macro and Microchromosome Probes.

PLOS ONE

Dear Dr. de Oliveira,

Thank you for submitting your manuscript to PLOS ONE. After careful consideration, we feel that it has merit but does not fully meet PLOS ONE’s publication criteria as it currently stands. Therefore, we invite you to submit a revised version of the manuscript that addresses the points raised during the review process.

As can you see below, your paper was revised by three reviewers. After a careful reading of the manuscript, and the reviewers’ suggestions, my decision is a major revision of your paper. I recommend that you answer all the reviewers’ questions in detail. (Note that reviewer 2 has included an attachment with their comments).

All reviewers suggest that the introduction should be more informative, and I agree with them. I also agree that the short title is too long and should be reduced.

Please, I would like you to know that the final acceptance of your manuscript will depend on the quality of the review of your manuscript and the responses to the reviewers' comments. Please let me know if you have any questions.

We look forward to receiving your revised manuscript.

Kind regards,

Maykon Passos Cristiano, D. Sc.

Academic Editor

PLOS ONE

2. Please provide additional details regarding participant consent. In the Methods section, please ensure that you have specified (1) whether consent was informed and (2) what type you obtained (for instance, written or verbal). If your study included minors, state whether you obtained consent from parents or guardians. If the need for consent was waived by the ethics committee, please include this information.

“This research was partially funded by a grant to EHCO from CNPq (307382/2019-2) and to MAFS from the Wellcome Trust in support of the Cambridge Resource Centre for Comparative Genomics and by the Biotechnology and Biological Sciences Research Council (BB/K008161/1) to the University of Kent.” 

Please include this amended Role of Funder statement in your cover letter; we will change the online submission form on your behalf."

5. Thank you for stating the following in the Financial Section of your manuscript:

“This research was partially funded by a grant to EHCO from CNPq (307382/2019-2) and to MAFS from the Wellcome Trust in support of the Cambridge Resource Centre for Comparative Genomics and by the Biotechnology and Biological Sciences Research Council (BB/K008161/1) to the University of Kent.”

Funding information should not appear in the Financial section or other areas of your manuscript. We will only publish funding information present in the Funding Statement section of the online submission form.

“This research was partially funded by a grant to EHCO from CNPq (307382/2019-2) and to MAFS from the Wellcome Trust in support of the Cambridge Resource Centre for Comparative Genomics and by the Biotechnology and Biological Sciences Research Council (BB/K008161/1) to the University of Kent.”

Reviewers' comments:

Reviewer's Responses to Questions

**Comments to the Author**

1. Is the manuscript technically sound, and do the data support the conclusions?

Reviewer #1: Yes

Reviewer #2: Yes

Reviewer #3: Partly

2. Has the statistical analysis been performed appropriately and rigorously? 

Reviewer #1: N/A

Reviewer #2: N/A

Reviewer #3: Yes

3. Have the authors made all data underlying the findings in their manuscript fully available?

Reviewer #1: Yes

Reviewer #2: Yes

Reviewer #3: Yes

4. Is the manuscript presented in an intelligible fashion and written in standard English?

Reviewer #1: Yes

Reviewer #2: Yes

Reviewer #3: Yes

5. Review Comments to the Author

Reviewer #1: The work from Carvalho et al brings and interesting analysis in the in the Black-Hawk-Eagle (Spizaetus tyrannus) using BAC and WCP experiments, both derived from Gallus gallus. The results are very well represented in high-quality figures and the and brings new pieces of information for the big puzzle that avian karyotype evolution represents. Below I indicate some points to be adjusted and put special attention to one suggestion for discussion: The work would profit too much if the analyses were done under a phylogenetic context. An updated phylogeny with the special focus on the phylogenetic position and relation between Chicken and Spizaetus would bring more light to the general results found.

Abstract

I missed some introduction about BIRDs cytogenetic in the beginning of Abstract. The authors go too direct about Accipitriformes and most readers do not have such phylogenetic information to which group we are dealing with.

Why are the authors aiming to investigate homologies with chicken? (Please see comments below). Abstract also brings some previous data on NNI that were not originated from this work. I suggest removing. The terms STY and NNI are used without any previous information/explanation about, causing misunderstandings.

A final conclusion is missing in the abstract.

Introduction

Page3 line3: Present an unusual (delete with)

Page 3 line 12: The accurate identification of the chromosomal pairs involved.… if we aim to identify synapomophies….

Introduction miss a clear motivation for the study. Why using GGA probes? I missed some info (Figure would be better) about the phylogenetic position and relation between Chicken and Spizaetus. Moreover, inform here as well thar STY corresponds to Spizaetus tyrannus on its first mention.

Results

Subtitle Karyotypes before the first paragraph

Figures

Figure 3: I did not understand why using the metaphases present in right column? The signals and chromosomal morphology are clearly visible in the FISH images.

Scale bars are missing in all figures.

Reviewer #2: In this study, the authors used comparative chromosome painting to reveal homology of chromosomal segments between Spizaetus tyrannus (the Black-Hawk-Eagle) and Gallus gallus. The study is conceptually and methodically motivated. The karyotype of S. tyrannus was previously studied only using conventional chromosome staining technique that showed 2n=68 (32m/sm+8st+18a+8m+ZW), and this happened for the first time that the karyotype of this bird of prey was studied by comparative chromosome painting. As a result, the study provides a novel insight into the cytogenetics of the Black-Hawk-Eagle. Both whole chromosome-specific G. gallus probes of the 1st-10th pairs and chromosome-specific G. gallus BAC probes from 11 pairs of microchromosomes were used. The study evidenced 29 evolutionary fission-fusion events that happened in the evolution of S. tyrannus and identified the particular chromosome pairs in the referenced G. gallus karyotype, which have undergone restructuring or have remained unchanged. Another sufficient result concerns the comparison between S. tyrannus and Nisaetus nipalensis orientalis, which is its only close relative studied so far by comparative chromosome painting. The comparison showed both similarities and differences between the species. The MS is well illustrated by tables and pictures of good quality.

In general, the work is interesting and deserves publication in the journal.

However, there are some shortcomings in the work. Main disadvantages are (1) almost complete absence of basic data on the karyotypes of the discussed species, which makes it difficult to adequately assess the results obtained, and (2) ignorance of the taxonomic component.

All other comments that I have are mainly suggestions for improving the article (unfortunately, there is no line numbering in the MS, which makes it difficult for the reviewer to work).

1. Running title. It is too long, should be shorter, e.g., “Comparative Chromosome Painting in Spizaetus tyrannus and Gallus gallus“.

2. Abstract. Please, decipher GGA, should be Gallus gallus (GGA), as done above for Spizaetus tyrannus (STY)

3. Introduction.

Page 3. Paragraph 1. Please, provide a putative avian ancestral karyotype, with 2n and the number of microchromosomes.

Page 3. Paragraph 1. Here (or further, in Discussion), provide the G. gallus karyotype to make further reasoning clearer.

Page 3. Paragraph 3. Unify way of citing – in other places you use numbering.

Page 4. Paragraph 4. Please, specify the species studied, Spizaetus tyrannus

4. Discussion.

Page 6, Paragraph 1. Please, expand the statement “We report slight differences in chromosome morphology….” by clarifying what Tagliarini et al. (2007) have reported. Describe the differences, give for comparison one and the other karyotypes (otherwise, you only have a declaration).

Page 7. Paragraph 3. The Japanese mountain hawk-eagle Nisaetus nipalensis orientalis (Temminck & Schlegel, 1844), not the Hodgson's hawk-eagle Nisaetus nipalensis Hodgson, 1836.

Other remarks can be found in the MS attached.

Reviewer #3: This study provides a cytogenetic mapping of the Black-Hawk-Eagle (Spizaetus tyrannus) using whole-chromosome paints and BAC probes of Gallus gallus. Using this cytogenetic mapping, the authors investigate the chromosome homologies between Spizaetus tyrannus and Gallus gallus. Overall the manuscript presents an advance in cytogenetic of Accipitriformes. The manuscript is written in understandable English, but some procedures require more details. Please see below my

comments that could help to further enhance the quality of the study.

1) This version is without line numbers to make it easier for reviewers to comment on the text.

2) Abstract: “chicken (Gallus gallus – GGA)”.

3) Introduction and discussion: The manuscript is written in a manner suitable for a more specialized journal such as Cytogenetic and Genome Research or Comparative Cytogenetics, but not ideally for a general journal such as PlosOne. The introduction and discussion are rather limited to the focus of the analyses. This is a shame as I think the authors did a good job to gather nice cytogenetic results. Thus, the authors should consider redrafting these sections on a broad context.

The introduction could be more informative about the study as a whole. For example, the authors can give information about the contribution of this kind of work to karyotypic evolution and chromosomal organization of Accipitriformes. In this line, the last paragraph of introduction can be better written showing the importance of this study, and not merely comment “This study presents the cytogenetic mapping of a species…”. Moreover, other intriguing topic is about the origin of microchromosomes. And maybe the authors could comment a little bit about this topic in the introduction and also discussion. Below is a reference related to this topic.

- Waters, P. D., Patel, H. R., Ruíz-Herrera, A., Álvarez-González, L., Lister, N. C., Simakov, O., ... & Graves, J. A. M. (2021). Microchromosomes are building blocks of bird, reptile and mammal chromosomes. bioRxiv.

4) Results, paragraph 1: The authors commented that they detected “four pairs of microchromosomes” but they indicated five (29, 30, 31, 32, and 33). Considering the Figure 1B the pair 29 apparently is not a microchromosome.

5) Results, paragraph 2: “The most extreme examples are the fission of GGA1 into six pairs in STY, and GGA3 into three distinct pairs”. It seems that for GGA3 are four pairs (13, 16, 19, and 20) (Figure 4), right?

6) Discussion, paragraph 3: “…karyotype of STY when compared to Gallus”; “…karyotype of STY when compared to Gallus gallus”.

7) Discussion, paragraph 4: “…microchromosomes (STY4, 7 and 9; NNI2, 4 and 9), none of them”; “microchromosomes (STY: pairs 4, 7 and 9; NNI: pairs 2, 4 and 9), none of them”.

“…with microchromosomes (GGA: pairs 18, 19, 24 and 25)”.

8) Discussion, paragraph 5: “These results show that they are morphologically similar species that until the last decade were part of the same genus [14].”. I think the authors results did not show an association of chromosomal data with morphological similarity and this should be corrected.

9) Conclusion: “…of S.tyrannus should be…”; “…of S. tyrannus should be…”.

10) Methods, paragraph 1: What the name and country of the Zoos?

11) Methods, paragraph 2: “…and labelled directly by FTIC”; “and labelled directly by fluorescein isothiocyanate (FITC)”.

12) Methods: the results of this study are based on FISH with GGA probes. However, FISH procedure is not sufficiently described. I would appreciate if at least main/important steps of the FISH procedure are described. This would also avoid any doubt about the accuracy of the results.

13) Figure 3: “FITC”.

14) Please, give a scale bar information for all figures. It is very important, specially considering the presence of macrochromosomes and microchromosomes.

6. PLOS authors have the option to publish the peer review history of their article (what does this mean?). If published, this will include your full peer review and any attached files.

Reviewer #1: No

Reviewer #2: No

Reviewer #3: No

---

## [Author Response · Author response to Decision Letter 0]

17 Sep 2021

Dear Editor and reviewer

We are thankful for the constructive reviews that we received; they certainly helped us

to improve the manuscript. Please find below our responses to each of your comments.

Sincerely,

Reviewer #1: 

1- The work from Carvalho et al brings and interesting analysis in the in the Black-Hawk-Eagle (Spizaetus tyrannus) using BAC and WCP experiments, both derived from Gallus gallus. The results are very well represented in high-quality figures and the and brings new pieces of information for the big puzzle that avian karyotype evolution represents. Below I indicate some points to be adjusted and put special attention to one suggestion for discussion: The work would profit too much if the analyses were done under a phylogenetic context. An updated phylogeny with the special focus on the phylogenetic position and relation between Chicken and Spizaetus would bring more light to the general results found.

A- From the cytotaxonomic point of view, Gallus gallus has a basal karyotype, very similar to the avian putative ancestor. As this species is an important biological model, and because it also retained a plesiomorphic karyotype, GGA probes are used as a standard for chromosomal studies in birds. However, except for the use of other probes, which could reveal intrachromosomal rearrangements, a phylogenetic analysis of GGA and birds of prey is not very informative, except as outgroup 

Abstract

2- I missed some introduction about BIRDs cytogenetic in the beginning of Abstract. The authors go too direct about Accipitriformes and most readers do not have such phylogenetic information to which group we are dealing with.

A- We agree, and added an introduction to the abstract “Although most birds show karyotypes with diploid number around 2n=80, with few macrochromosomes and many microchromosomes pairs, some groups, such as the Accipitriformes, are characterized by a large karyotypic reorganization, which resulted in complements with low diploid numbers, and a smaller number of microchromosomal pairs when compared to other birds”

3- Why are the authors aiming to investigate homologies with chicken? 

A- Because this species is an important biological model, and because it also retained a plesiomorphic karyotype, GGA probes are used as a standard for chromosomal studies in birds. We added a small explanation in the introduction “In general, studies focusing on chromosome evolution in birds are based in comparative chromosome painting using chicken whole chromosome probes (Gallus gallus – GGA, 2n=78), due to the similarity of the karyotype of this species with the PAK [5].”

4- Abstract also brings some previous data on NNI that were not originated from this work. I suggest removing.

A- We removed it 

5- The terms STY and NNI are used without any previous information/explanation about, causing misunderstandings.

A- We corrected it

6- A final conclusion is missing in the abstract.

A- We added a conclusion in the abstract: “Comparison with hybridization pattern of the Japanese-Mountain-Eagle, the only Aquilinae analyzed by comparative chromosome painting previously, did not reveal any synapomorphy that could represent a chromosome signature to this subfamily. Therefore, conclusions about karyotype evolution in Aquilinae require additional painting studies”. 

Introduction

7- Page3 line3: Present an unusual (delete with)

A- We deleted it.

8- Page 3 line 12: The accurate identification of the chromosomal pairs involved.… if we aim to identify synapomophies….

A- We corrected it

9- Introduction miss a clear motivation for the study. 

A- We improved it: “Among them, the subfamily Aquilinae includes medium and large species, distributed globally, usually known as booted eagle. Usually, ten genera are found within Aquilinae. Cytogenetically, the only information concerning Aquilinae is the definition of the diploid number of six species (four genera), ranging from 2n=66 to 82 [19]”. 

10- Why using GGA probes? 

A- Because this species is an important biological model, and because it also retained a plesiomorphic karyotype, GGA probes are used as a standard for chromosomal studies in birds. We added a small explanation in the introduction. 

11- I missed some info (Figure would be better) about the phylogenetic position and relation between Chicken and Spizaetus. 

A- The authors agree that for this study are not necessary introduce some information about the phylogenetic position between Gallus gallus and S. tyrannus, however short information of phylogenetic relationship of the S.tyrannus within the Aquilinae Subfamily were added in the introduction aiming improve the reading of this paper.

12- Moreover, inform here as well thar STY corresponds to Spizaetus tyrannus on its first mention.

A- This was corrected.

Results

13- Subtitle Karyotypes before the first paragraph.

A- We added it.

Figures

14- Figure 3: I did not understand why using the metaphases present in right column? The signals and chromosomal morphology are clearly visible in the FISH images. Scale bars are missing in all figures.

A- We removed the right column and added the scale bars in all figures.

Reviewer #2:

 In this study, the authors used comparative chromosome painting to reveal homology of chromosomal segments between Spizaetus tyrannus (the Black-Hawk-Eagle) and Gallus gallus. The study is conceptually and methodically motivated. The karyotype of S. tyrannus was previously studied only using conventional chromosome staining technique that showed 2n=68 (32m/sm+8st+18a+8m+ZW), and this happened for the first time that the karyotype of this bird of prey was studied by comparative chromosome painting. As a result, the study provides a novel insight into the cytogenetics of the Black-Hawk-Eagle. Both whole chromosome-specific G. gallus probes of the 1st-10th pairs and chromosome-specific G. gallus BAC probes from 11 pairs of microchromosomes were used. The study evidenced 29 evolutionary fission-fusion events that happened in the evolution of S. tyrannus and identified the particular chromosome pairs in the referenced G. gallus karyotype, which have undergone restructuring or have remained unchanged. Another sufficient result concerns the comparison between S. tyrannus and Nisaetus nipalensis orientalis, which is its only close relative studied so far by comparative chromosome painting. The comparison showed both similarities and differences between the species. The MS is well illustrated by tables and pictures of good quality.

In general, the work is interesting and deserves publication in the journal.

However, there are some shortcomings in the work. Main disadvantages are (1) almost complete absence of basic data on the karyotypes of the discussed species, which makes it difficult to adequately assess the results obtained, and (2) ignorance of the taxonomic component.

A- We try to improve it.

All other comments that I have are mainly suggestions for improving the article (unfortunately, there is no line numbering in the MS, which makes it difficult for the reviewer to work).

1-Running title. It is too long, should be shorter, e.g., “Comparative Chromosome Painting in Spizaetus tyrannus and Gallus gallus“.

A- W corrected it.

2. Abstract. Please, decipher GGA, should be Gallus gallus (GGA), as done above for Spizaetus tyrannus (STY)

A- We corrected it.

Introduction.

3- Page 3. Paragraph 1. Please, provide a putative avian ancestral karyotype, with 2n and the number of microchromosomes.

A- We provided this information in the introduction of the paper.

4- Page 3. Paragraph 1. Here (or further, in Discussion), provide the G. gallus karyotype to make further reasoning clearer.

A- We did it.

5- Page 3. Paragraph 3. Unify way of citing – in other places you use numbering.

A- We corrected it.

6- Page 4. Paragraph 4. Please, specify the species studied, Spizaetus tyrannus

A- We corrected it.

Discussion.

7- Page 6, Paragraph 1. Please, expand the statement “We report slight differences in chromosome morphology….” by clarifying what Tagliarini et al. (2007) have reported. Describe the differences, give for comparison one and the other karyotypes (otherwise, you only have a declaration).

A- In order to clarify the difference in chromosome morphology described between this study and the Tagliarini et al., (2007), we made a comparative table that was introduce in the results.

8- Page 7. Paragraph 3. The Japanese mountain hawk-eagle Nisaetus nipalensis orientalis (Temminck & Schlegel, 1844), not the Hodgson's hawk-eagle Nisaetus nipalensis Hodgson, 1836.

A- We corrected it.

9- Other remarks can be found in the MS attached.

A- We followed all the instructions of attached.

Reviewer #3: 

This study provides a cytogenetic mapping of the Black-Hawk-Eagle (Spizaetus tyrannus) using whole-chromosome paints and BAC probes of Gallus gallus. Using this cytogenetic mapping, the authors investigate the chromosome homologies between Spizaetus tyrannus and Gallus gallus. Overall the manuscript presents an advance in cytogenetic of Accipitriformes. The manuscript is written in understandable English, but some procedures require more details. Please see below my

comments that could help to further enhance the quality of the study.

2) Abstract: “chicken (Gallus gallus – GGA)”.

A- We corrected it.

3) Introduction and discussion: The manuscript is written in a manner suitable for a more specialized journal such as Cytogenetic and Genome Research or Comparative Cytogenetics, but not ideally for a general journal such as PlosOne. The introduction and discussion are rather limited to the focus of the analyses. This is a shame as I think the authors did a good job to gather nice cytogenetic results. Thus, the authors should consider redrafting these sections on a broad context.

The introduction could be more informative about the study as a whole. For example, the authors can give information about the contribution of this kind of work to karyotypic evolution and chromosomal organization of Accipitriformes. In this line, the last paragraph of introduction can be better written showing the importance of this study, and not merely comment “This study presents the cytogenetic mapping of a species…”. Moreover, other intriguing topic is about the origin of microchromosomes. And maybe the authors could comment a little bit about this topic in the introduction and also discussion. Below is a reference related to this topic.

- Waters, P. D., Patel, H. R., Ruíz-Herrera, A., Álvarez-González, L., Lister, N. C., Simakov, O., ... & Graves, J. A. M. (2021). Microchromosomes are building blocks of bird, reptile and mammal chromosomes. bioRxiv.

A- The introduction was rewritten and several new information were added to improve the understanding of the manuscript. Also, some information from the reference recommended by the reviewer were considerate in this topic.

4) Results, paragraph 1: The authors commented that they detected “four pairs of microchromosomes” but they indicated five (29, 30, 31, 32, and 33). Considering the Figure 1B the pair 29 apparently is not a microchromosome.

A- We reviewed the Karyotype of S.tyrannus and considered only four pairs of microchromosomes ( Pairs: 30, 31, 32 and 33). Also, we corrected it throughout of the text. 

5) Results, paragraph 2: “The most extreme examples are the fission of GGA1 into six pairs in STY, and GGA3 into three distinct pairs”. It seems that for GGA3 are four pairs (13, 16, 19, and 20) (Figure 4), right?

A- The reviewer is right. The GGA3 probe correspond to four pairs (13,16,19 and 20).

We changed it. 

6) Discussion, paragraph 3: “…karyotype of STY when compared to Gallus”; “…karyotype of STY when compared to Gallus gallus”.

A- We corrected it.

7) Discussion, paragraph 4: “…microchromosomes (STY4, 7 and 9; NNI2, 4 and 9), none of them”; “microchromosomes (STY: pairs 4, 7 and 9; NNI: pairs 2, 4 and 9), none of them”.

“…with microchromosomes (GGA: pairs 18, 19, 24 and 25)”.

A- We corrected it.

8) Discussion, paragraph 5: “These results show that they are morphologically similar species that until the last decade were part of the same genus [14].”. I think the authors results did not show an association of chromosomal data with morphological similarity and this should be corrected.

A- We removed it.

9) Conclusion: “…of S.tyrannus should be…”; “…of S. tyrannus should be…”.

A- We corrected it.

10) Methods, paragraph 1: What the name and country of the Zoos?

A- The information was added in the Methods.

11) Methods, paragraph 2: “…and labelled directly by FTIC”; “and labelled directly by fluorescein isothiocyanate (FITC)”.

A- We changed it.

12) Methods: the results of this study are based on FISH with GGA probes. However, FISH procedure is not sufficiently described. I would appreciate if at least main/important steps of the FISH procedure are described. This would also avoid any doubt about the accuracy of the results.

A- We provided the main steps of the FISH procedure.

13) Figure 3: “FITC”.

A- We corrected it.

14) Please, give a scale bar information for all figures. It is very important, specially considering the presence of macrochromosomes and microchromosomes.

A- We added the scale bar in all figures.

---

## [Decision Letter · Decision Letter 1]

27 Oct 2021

PONE-D-21-20639R1Comparative Chromosome Painting in the Black-Hawk-Eagle (Spizaetus tyrannus) and Gallus gallus with the use of Macro and Microchromosome Probes.PLOS ONE

Dear Dr. de Oliveira,

Thank you for submitting your manuscript to PLOS ONE. After careful consideration, we feel that it has merit but does not fully meet PLOS ONE’s publication criteria as it currently stands. Therefore, we invite you to submit a revised version of the manuscript that addresses the points raised during the review process.

We look forward to receiving your revised manuscript.

Kind regards,

Maykon Passos Cristiano, D. Sc.

Academic Editor

PLOS ONE

Journal Requirements:

Additional Editor Comments:

I return with another round of review. Two reviewers (2 and 3) make small suggestions to improve the quality of the manuscript. Also, the Plos one does not submit proof before publication, for this reason, I suggest special care in this review.

Reviewers' comments:

Reviewer's Responses to Questions

**Comments to the Author**

1. If the authors have adequately addressed your comments raised in a previous round of review and you feel that this manuscript is now acceptable for publication, you may indicate that here to bypass the “Comments to the Author” section, enter your conflict of interest statement in the “Confidential to Editor” section, and submit your "Accept" recommendation.

Reviewer #1: All comments have been addressed

Reviewer #2: All comments have been addressed

Reviewer #3: All comments have been addressed

2. Is the manuscript technically sound, and do the data support the conclusions?

Reviewer #1: Yes

Reviewer #2: Yes

Reviewer #3: Yes

3. Has the statistical analysis been performed appropriately and rigorously? 

Reviewer #1: Yes

Reviewer #2: N/A

Reviewer #3: N/A

4. Have the authors made all data underlying the findings in their manuscript fully available?

Reviewer #1: Yes

Reviewer #2: Yes

Reviewer #3: Yes

5. Is the manuscript presented in an intelligible fashion and written in standard English?

Reviewer #1: Yes

Reviewer #2: Yes

Reviewer #3: Yes

6. Review Comments to the Author

Reviewer #1: (No Response)

Reviewer #2: The manuscript has been extensively revised. Generally, all my comments were addressed and I am satisfied with the revision. However, minor typos remain (listed below and marked in the attached PDF), which can be corrected while editorial processing and preparing the manuscript for publication.

Title

1. Comparative Chromosome Painting in the Black-Hawk-Eagle

(Spizaetus tyrannus) and Gallus gallus with the use of Macro

and Microchromosome Probes.

It is better to use only Latin names in both cases, e.g.

Comparative Chromosome Painting in Spizaetus tyrannus and Gallus gallus with the use of Macro-and Microchromosome Probes.

Then, Macro-and Microchromosome Probes should be written with a hyphen.

Abstract

1. Here and elsewhere (Introduction, Results), should not be duplicated diploid number and 2n, as for example in the case “diploid number around 2n=80”. Should be either diploid number around 80 or 2n around 80

2. The family Accipitridae contains 14 subfamilies, not only Aquilinae, then, the phrase “the Accipitridae family is the most diverse and includes the subfamily Aquilinae…” needs to be revised (see my suggestion in the PDF text).

Results

1. Table 3. Please, explain in the title the abbreviation “q”

-

Reviewer #3: Carvalho et al. 2021 – Resubmission “Comparative Chromosome Painting in the Black-Hawk-Eagle (Spizaetus tyrannus) and Gallus gallus with the use of Macro and Microchromosome Probes”. This is a re-review of the manuscript for PlosOne. The new version of the manuscript is improved, and I do think the results overall are publishable. I only have a few suggestions that can be considered during the production process.

- Please alphabetic order: “…In fact, rearrangements involving microchromosomes were detected in few orders: Psittaciformes, Cuculiformes, Suliformes, Caprimulgiformes and the Accipitriformes [13-15].”

-“Gallus gallus probes used in the fluorescent in situ hybridization (FISH) experiments produced…”.

7. PLOS authors have the option to publish the peer review history of their article (what does this mean?). If published, this will include your full peer review and any attached files.

Reviewer #1: No

Reviewer #2: No

Reviewer #3: No

---

## [Author Response · Author response to Decision Letter 1]

27 Oct 2021

Reviewer #2: The manuscript has been extensively revised. Generally, all my comments were addressed and I am satisfied with the revision. However, minor typos remain (listed below and marked in the attached PDF), which can be corrected while editorial processing and preparing the manuscript for publication.

Answer: Thank you very much for the positive feedback, we accepted all the suggestion in the PDF file.

Title

1. Comparative Chromosome Painting in the Black-Hawk-Eagle (Spizaetus tyrannus) and Gallus gallus with the use of Macro and Microchromosome Probes. It is better to use only Latin names in both cases, e.g. Comparative Chromosome Painting in Spizaetus tyrannus and Gallus gallus with the use of Macro-and Microchromosome Probes. Then, Macro-and Microchromosome Probes should be written with a hyphen.

Answer: Thank you, we accepted the suggestion.

Abstract

1. Here and elsewhere (Introduction, Results), should not be duplicated diploid number and 2n, as for example in the case “diploid number around 2n=80”. Should be either diploid number around 80 or 2n around 80

Answer: We corrected it.

2. The family Accipitridae contains 14 subfamilies, not only Aquilinae, then, the phrase “the Accipitridae family is the most diverse and includes the subfamily Aquilinae…” needs to be revised (see my suggestion in the PDF text).

Answer: We corrected it.

Results

1. Table 3. Please, explain in the title the abbreviation “q”.

Answer: We explained it.

Reviewer #3: Carvalho et al. 2021 – Resubmission “Comparative Chromosome Painting in the Black-Hawk-Eagle (Spizaetus tyrannus) and Gallus gallus with the use of Macro and Microchromosome Probes”. This is a re-review of the manuscript for PlosOne. The new version of the manuscript is improved, and I do think the results overall are publishable. I only have a few suggestions that can be considered during the production process.

Answer: Thank you very much for the positive feedback.

- Please alphabetic order: “…In fact, rearrangements involving microchromosomes were detected in few orders: Psittaciformes, Cuculiformes, Suliformes, Caprimulgiformes and the Accipitriformes [13-15].”

Answer: We corrected it.

-“Gallus gallus probes used in the fluorescent in situ hybridization (FISH) experiments produced…”.

Answer: We corrected it.

---

## [Editor Report · Decision Letter 2]

29 Oct 2021

Comparative Chromosome Painting in Spizaetus tyrannus) and Gallus gallus with the use of Macro and Microchromosome Probes.

PONE-D-21-20639R2

Dear Dr. de Oliveira,

We’re pleased to inform you that your manuscript has been judged scientifically suitable for publication and will be formally accepted for publication once it meets all outstanding technical requirements.

Kind regards,

Maykon Passos Cristiano, D. Sc.

Academic Editor

PLOS ONE

---

## [Editor Report · Acceptance letter]

5 Nov 2021

PONE-D-21-20639R2 

Comparative Chromosome Painting in *Spizaetus tyrannus* and *Gallus gallus* with the use of Macro- and Microchromosome Probes 

Dear Dr. de Oliveira:

I'm pleased to inform you that your manuscript has been deemed suitable for publication in PLOS ONE. Congratulations! Your manuscript is now with our production department. 

Kind regards, 

on behalf of

Mr. Maykon Passos Cristiano 

Academic Editor

PLOS ONE